# Dissociating Language and Thought in Human Reasoning [note 1]

**DOI:** 10.3390/brainsci13010067

**Published:** 2022-12-29

**Authors:** John P. Coetzee, Micah A. Johnson, Youngzie Lee, Allan D. Wu, Marco Iacoboni, Martin M. Monti

**Affiliations:** 1Department of Psychiatry and Behavioral Sciences, Stanford University School of Medicine, 401 Quarry Road, Stanford, CA 94305, USA; 2VA Palo Alto Health Care System, Polytrauma Division, 3801 Miranda Avenue, Palo Alto, CA 94304, USA; 3Department of Psychology, University of California Los Angeles, Los Angeles, CA 90095, USA; 4Department of Neurology, David Geffen School of Medicine, University of California Los Angeles, Los Angeles, CA 90095, USA; 5Brain Research Institute (BRI), University of California Los Angeles, Los Angeles, CA 90095, USA; 6Ahmanson-Lovelace Brain Mapping Center, University of California Los Angeles, Los Angeles, CA 90095, USA; 7Department of Psychiatry and Biobehavioral Sciences, Semel Institute for Neuroscience and Human Behavior, David Geffen School of Medicine, University of California Los Angeles, Los Angeles, CA 90095, USA; 8Brain Injury Research Center (BIRC), Department of Neurosurgery, David Geffen School of Medicine at UCLA, Los Angeles, CA 90095, USA

**Keywords:** language, cognition, deductive reasoning, neuromodulation, theta burst stimulation, transcranial magnetic stimulation

## Abstract

What is the relationship between language and complex thought? In the context of deductive reasoning there are two main views. Under the first, which we label here the language-centric view, language is central to the syntax-like combinatorial operations of complex reasoning. Under the second, which we label here the language-independent view, these operations are dissociable from the mechanisms of natural language. We applied continuous theta burst stimulation (cTBS), a form of noninvasive neuromodulation, to healthy adult participants to transiently inhibit a subregion of Broca’s area (left BA44) associated in prior work with parsing the syntactic relations of natural language. We similarly inhibited a subregion of dorsomedial frontal cortex (left medial BA8) which has been associated with core features of logical reasoning. There was a significant interaction between task and stimulation site. Post hoc tests revealed that performance on a linguistic reasoning task, but not deductive reasoning task, was significantly impaired after inhibition of left BA44, and performance on a deductive reasoning task, but not linguistic reasoning task, was decreased after inhibition of left medial BA8 (however not significantly). Subsequent linear contrasts supported this pattern. These novel results suggest that deductive reasoning may be dissociable from linguistic processes in the adult human brain, consistent with the language-independent view.

## 1. Dissociating Language and Thought in Human Reasoning

Does language shape human cognition? There are two dominant perspectives on this question [1,2,3,4,5]. The first is the *cognitive conception of language*, in which language is seen as being constitutively involved in human cognition and as forming, at least partially, the medium of thought [3,5,6,7,8]. At its extreme, this view proposes that some types of thought cannot be entertained in the absence of language [9]. The second is the *communicative conception of language*, in which language is seen primarily as an inert conduit for the communication of preexisting (i.e., non-linguistic) mental representations from one mind to another through a mutually intelligible code [10,11,12].

The antagonism between these two polar positions has long been investigated in the context of a number of cognitive domains including the development of ontological categories [13], color perception [14,15], spatial cognition and geometry [16,17], number cognition [18,19], action representation [20], music cognition [21,22], and theory of mind [23,24], among many others. More recently, this debate has leveraged the tools of neuroscience to investigate the degree to which the neural substrate of natural language also participates in other aspects of human cognition [11]. In particular, it has been proposed that a segment of the left inferior frontal gyrus (IFG; often referred to as Broca’s Area), which has long been associated with processing the hierarchical sequences of natural language [25,26,27,28,29], might actually function supramodally as a processor of hierarchical sequences across domains of human thought [30,31,32].

In the specific context of human reasoning, it has long been debated what role language plays in deductive inference-making [33,34,35,36,37]. A growing body of neuroimaging work has renewed debate between two positions on this question [38,39,40,41,42,43,44,45]. Under the first view, which we will call here the language centric-view, the syntax-like operations of deductive reasoning are mainly based upon the neural mechanisms of language in the left IFG [46,47], and thus best understood as linguistic in nature. This view rests, neuroscientifically, on a number of publications showing activation of the left IFG, as detected with functional MRI, during logic inferences [46,47,48,49,50,51]. Reverberi and colleagues [46], for example, used an event-related fMRI approach to probe different cognitive components of the inferential process and found that simple conditional and disjunctive problems (i.e., based on logic connectives such as “if … then”, “or”) recruited, at the moment of inference, the left IFG (in left Brodmann’s Area 44 (LBA44)) along with other lateral frontal (i.e., BA6) and parietal (BA40) regions. Furthermore, these findings were shown to apply not only to conditional and disjunctive propositional problems but also to syllogistic inferences (i.e., based on quantifiers such as “all”, “some”, “none”) [47]. Indeed, Reverberi and colleagues also showed that LBA44 and LBA45 in the IFG were recruited during both the encoding and premise integration stages of a deductive inference, which was interpreted by the authors as suggesting a central role for these primarily linguistic regions in deductive reasoning (See [42,44] for a review).

Under the second view, which we will call here the language-independent view, deductive reasoning is mainly supported by neural mechanisms that extend beyond the areas of the brain conventionally associated with processing the hierarchical relationships of language [52], spanning left dorsomedial frontal and frontopolar cortices (in Brodmann areas (BA) 8 and 10, respectively) [39,40,53,54,55,56,57,58,59,60]. This view is also primarily supported, in the neuroscientific literature, by functional MRI work. Coetzee et al. [39], for example, used an event-related fMRI approach to probe neural activation during a conditional reasoning task and found that areas in the left IFG were selectively activated by increased (non-logic) verbal processing demands, but not by increased deductive reasoning demands. Conversely, medial frontal cortex was specifically activated by increased deductive reasoning load, but not by verbal non-deductive load. This study was in part a replication of earlier work by Monti et al. [53], in which the comparison of brain activations during difficult versus easy deductive inferences highlighted a non-linguistic network subserving deductive reasoning, interpreted to be composed of core deductive regions (in frontopolar BA10 and frontomedial BA8) and general cognitive support regions (including left frontal BA6 & 47, and parietal (BA7 & 40)) that are more flexibly recruited depending on working memory and other ancillary cognitive processes. Similar findings were reported in a decidedly different study design by Rodriguez-Moreno et al. [54]. In this work, Rodriguez-Moreno and colleagues found, using a conjunction analysis, that syllogisms presented in either a visual or auditory manner revealed a similar pattern of brain regions to the studies mentioned above, with a similar absence of recruitment of classical language areas.

Under the language-independent view, linguistic resources are understood as necessary to decode verbally presented logic statements into mental representations. However, beyond this initial stage, the linguistic computations implemented in the left IFG are considered to play little to no role in the mental operations of deductive inference-making [42,53,61]. It should be noted that this hypothesis is perfectly compatible with the idea that heuristic reasoning, as in the case of belief bias, might well be exerted through language [62] since they are not, themselves, logic processes.

In order to test these two perspectives, we utilized the capabilities of noninvasive brain stimulation. Unlike functional magnetic resonance imaging (fMRI) or electroencephalography (EEG), which are correlational in nature, brain stimulation approaches make it possible to directly manipulate cortical excitability in a localized fashion, enabling causal inferences about the relationships between specific brain circuits, cognitive processes, and behaviors [63,64]. We used a transcranial magnetic stimulation (TMS) device with a continuous theta burst stimulation (cTBS) protocol, an approach that is capable of transiently inhibiting or reducing neural activity in cortical areas and their participating networks [65,66,67,68]. We note here that TMS has already been used in prior literature to investigate the syntactic properties of Broca’s area [69,70,71] and the neural bases of deductive reasoning [72,73,74]. The present study, however, is the first to explicitly test, side by side, the effects of inhibiting the neural processes encoded in Broca’s area on the syntactic operations of language and on the syntax-like operations of deductive reasoning.

Our experimental design involved two tasks of interest, linguistic reasoning and deductive reasoning, both adapted from prior work [52] (see Figure 1 and Table 1), and two target brain sites of interest, left BA44 (LBA44) and a left hemisphere location in medial BA8 (MBA8) (see Figure 2), previously associated with linguistic processing [25,26,27,28,29] and deductive inference [39,52,53,54], respectively. In addition, our design included a control task, grammaticality judgments, also adapted from prior work [52] (see Table 1), and a control site, in the left transverse occipital sulcus (LTOS), previously associated with visual scene processing [75,76,77] and chosen for its anatomical and functional distance from the two target sites of interest (see Figure 2).

This fully within-subjects paradigm was conceived to test three very specific hypotheses derived from over 20 years of functional imaging of deductive reasoning [39,43,46,47,50,52,53,54,78]. First, if the neural mechanisms of the syntax-like operations of deductive inference are dissociable from the syntactic operations of natural language, we ought to observe a significant interaction between the two factors of our design (i.e., task and target site). Second, if the language-independent view of deduction is correct [39,52,53], cTBS inhibition of LBA44 ought to decrease performance on the linguistic reasoning task while sparing performance on the deductive reasoning task (i.e., we expect a significant difference in post-cTBS performance across the two tasks, with a specific directionality). Conversely, if the language-centric view of deduction is correct [46,47], cTBS inhibition of LBA44 ought to result in impaired performance on both reasoning tasks. Third, if MBA8 is indeed a core node for deductive inference [42,52], cTBS inhibition to this region ought to impair performance on the deductive reasoning task but not on the linguistic reasoning task (i.e., we expect a significant difference in post-cTBS performance across the two tasks, with a specific directionality).

## 2. Methods

### 2.1. Participants

A total of 45 cTBS sessions were obtained from 15 participants in a three-session within-subjects design. All study procedures took place at the University of California, Los Angeles (UCLA). A power analysis was conducted with G*Power [79] using an effect size from a factorial ANOVA interaction in a previous study [73]. In that study by Tsujii et al., inhibitory repetitive TMS (rTMS) using a frequency of 1 Hz applied to the left IFG (LBA45) impaired reasoning about congruent but not incongruent syllogisms, thereby eliminating the belief bias, which is the bias that reasoners have for judging syllogisms as valid when the conclusion agrees with their preexisting knowledge. Tsujii et al. reported a conventionally “large” effect size (*η*^2^*p* = 0.13) for the effect of IFG stimulation on congruent reasoning. We further used an assumed α = 0.05, and determined that 15 participants would be needed to achieve 85% power to detect an interaction using a repeated measures ANOVA. This sample size is consistent with other previously published rTMS work involving the role of the IFG in cognition [80,81,82,83]. The mean age of the cohort in our study was 21.1 years old, ranging from 18 to 30 years old. Participants were recruited through flyers and from other (unrelated) studies. To be included, participants had to be right-handed, native English speakers, 18–50 years old, and have had no significant prior formal instruction in deductive reasoning (such as completing a symbolic logic course). In addition, we only selected participants who had a recent structural MRI available (from previous participation in a neuroimaging experiment at UCLA) to allow for MR-guided targeting with the TMS coil on the basis of individual brain anatomy (see below). In keeping with TMS safety standards [84], participants were excluded if they had metal implants in their head, regularly engaged in excessive alcohol use, were pregnant, had a family history of seizures, had been diagnosed with any significant medical, psychiatric or neurological conditions, or used any prescription medication that could lower their seizure threshold (i.e., bupropion). Participants were compensated $25 per hour for their time. Total compensation for each completing participant ranged from $125 to $175. The variability in compensation was usually a consequence of differing amounts of time needed to acquire a motor threshold.

### 2.2. Task and Stimuli

Task and stimuli materials (see Table 1) were adapted from prior studies [52,53]. For each of the three cTBS sessions, participants were presented with 156 stimuli in a visual format. Each stimulus consisted of an argument, defined as a set of two sentences presented one above a horizontal line and one below (see Table 1 and Figure 1). Half the arguments were “linguistic” in that they described a subject–object–patient relationship (i.e., “who did what to whom”; e.g., “Y gave X to Z.” and “X was given Z by Y.”). The remaining arguments were “logic” in that they described the logic implicature tying phrasal constituents together (i.e., “X, Y, Z”; e.g., “If Y or X then not Z.” and “If Z then not Y and not X.”). For each linguistic and logic argument, participants were asked to perform one of two tasks. In the reasoning task, they were asked to establish whether the two sentences of each argument matched in that they described logically equivalent states of affairs (that is, they had to decide whether the two sentences were transformations of one another). Half the arguments presented in the reasoning trials described the same state of affairs and half did not. In the grammaticality judgment task, by contrast, participants were merely asked to evaluate whether both sentences of each argument were grammatical (with no need to relate the two sentences to each other). Half the arguments presented in the grammaticality trials were grammatical and half were not. As done in other studies, ungrammatical arguments were obtained by altering word order in either sentence [33,61]. Half the ungrammatical sentences had an error in the sentence above the line, and half had the error in the sentence below the line. Overall, the 156 arguments that participants saw at each session included 104 reasoning trials (half with “linguistic” arguments and half with “logic arguments”) and 52 grammaticality judgment trials (also evenly divided between types of arguments). The reason for the smaller number of grammaticality trials is that the focus of the experiment was on logic reasoning and linguistic reasoning, while the grammaticality trials were used as merely a form of quality control.

It should be noted that, in the context of the reasoning task, linguistic and logic arguments emphasize different types of structure-dependent relationships [11]. When presented with linguistic arguments, the reasoning task required understanding the thematic relations of “X, Y, Z” with respect to the major verb of the sentence, across different syntactic constructs (e.g., X is a patient in “It was X that Y saw Z take.” but is an agent in “Z was seen by X taking Y.”). When presented with logic arguments, the reasoning task required understanding the logic relations tying phrasal constituents together across different statements (e.g., “If both X and Z then not Y.” and “If Y then either not X or not Z.”).

Importantly, participants were presented with varied exemplars of both types of arguments. For linguistic reasoning, exemplars used included: different grammatical structures (e.g., active [e.g., “Y gives X to Z.”]; passive [e.g., “Z was seen taking X by Y.”]; pronoun headed [e.g., “It was Z that was given to Y by X.”]; present tense [e.g., “W reports that Z tells X to Y.”]; future tense [e.g., “X will write that Z will be thought by Y to have said W.”]; relative clauses [e.g., “It was X that W knew was given to Y by Z.”]; among others); different verbs and different numbers of verbs (i.e., 1-verb exemplars: tell; see; give; 2-verb exemplars: think-say; tell-be; take-see; report-tell; give-be; 3-verb exemplars: report-tell-be; know-give-be; write-think-say; hear-see-take; report-see-take; think-be-say); and different numbers of thematic roles (i.e., 3 [e.g., “It is Y that is given by X to Z.”] and 4 [e.g., “Z was reported by W to have been seen taking Y by X.”]). A similar approach was applied to generating logic exemplars which featured different logic forms obtained by combining several logic operations (i.e., conditionals, conjunctions, dis-junctions, negations; employed in different combinations [e.g., “If either not Z or not X then either W or Y.”, “If both W and not X then either Y or not Z.”]) leading to the application of several logic rules (e.g., modus tollens, de Morgan), with three or four logic variables per argument (e.g., “If not X then both not Y and not Z.”, “If either not Z or not Y then both not W and not X.”). This variety of exemplars for each task was employed to ensure that our TMS interference approach, if successful, would be generalizable.

An integral component of the construction of our logic stimuli were de Morgan’s laws. These two transformational rules (described below) apply in both propositional logic and Boolean algebra, and are considered valid rules of inference [85,86]. These rules can be expressed as (1) not (A or B) = (not A) and (not B), and (2) not (A and B) = (not A) or (not B). Importantly, the “or” used in these rules when stated as such is an inclusive rather than an exclusive “or,” something of which we had to make our participants explicitly aware, at the screening task, given the ambiguity of the English connective term. In formal logic, these rules can be expressed as (1) *−(p∧q)*↔*−p∨−q*, and (2) *−(p∨q)*↔*−p∧−q.*

Our stimuli also included a relational complexity [87] manipulation by which, for each type of problem, half the trials included statements concerning the relationships between three variables (e.g., “X was given Y by Z.” and “If either X or Y then not Z.”) and the remainder included statements concerning the relationship between four variables (e.g., “W heard that Z was seen by Y taking X.” and “If either Z or W then both X and not Y.”). This manipulation was included in case cTBS turned out to have effects that depend on the amount of cognitive load. Since we found no significant main effect for the number of variables, and no significant two-way interaction with either site or task, and no significant three-way interaction (all *p* > 0.05; see discussion), we omit this variable from the reported results.

Overall, for each type of problem, half the arguments featured sentences describing the same state of affairs (i.e., where the two sentences match in the circumstance they describe). Assignment of the variables W, X, Y, Z to elements/phrasal constituents was randomized across arguments. In each session, the 156 arguments included 78 linguistic arguments and 78 logic arguments. For each type, 52 arguments were presented in reasoning trials, and 26 were presented in grammaticality judgment trials. Of the 156 trials, 36 (equally distributed across tasks) were presented prior to cTBS stimulation (i.e., baseline trials) and 120 (equally distributed across tasks) were presented after cTBS stimulation. The same 156 arguments were presented across the four sessions except for randomly allocating each argument to baseline or post-cTBS presentation and for different allocation of variables (i.e., W, X, Y, Z) to thematic roles/phrasal constituents. Within baseline and post-cTBS sequences, presentation order of each argument (and task) was randomized with the sole constraint that trials with identical parameters not occur consecutively.

As shown in Figure 1, each trial began with a one second fixation cross followed by a one second cue signaling to the participant whether they were to perform a reasoning task (with either linguistic or logic materials), cued by the word “MEANING”, or the grammaticality judgment task (with either linguistic or logic materials), cued by the word “GRAMMAR”. The cue was followed by on-screen presentation of the argument, with the two sentences arranged vertically, one above the other, separated by a horizontal line (cf., Figure 1). Given the randomized task order, a small “M” or “G” block letter at the top left of the screen served as a reminder of which tasks participants were expected to perform at each trial (as has been done in previous work [39]). Participants had up to a maximum of 15 s to press the A key for a positive answer (i.e., “the sentences describe the same state of affairs” and “both sentences are grammatical”, for the reasoning and grammaticality judgment task, respectively) and the L key for a negative answer (i.e., “the sentences do not describe the same state of affairs” and “one of the two sentences is grammatically incorrect”, for the reasoning and grammaticality judgment task, respectively). The trial terminated upon button-press or upon the elapsing of the allotted 15 s, after which a new trial would begin. Stimuli were delivered using Psychopy [88] on a Toshiba Satellite laptop running Windows 7.

### 2.3. Targets

We selected five cortical targets (two experimental, one control, and two for motor thresholding) for neurostimulation (see Figure 2) based on previous literature and initially defined on the Montreal Neurological Institute’s (MNI) standard T1 template. The first target was the pars opercularis subregion of Broca’s area in the left inferior frontal gyrus (LBA44; x = −50, y = 18, z = 18), which was identified from our previous fMRI study using a similar linguistic task [52]. The second target was a region in the dorsomedial frontal cortex (MBA8; x = −6, y = 40, z = 38) identified from our previous fMRI research as a hotspot for deductive inference based on a similar task [39,52]. The third target was a region in the lateral transverse occipital sulcus (LTOS; x = −25, y = −85, z = 25) [89], which we identified as a suitable control region likely unrelated to linguistic or deductive processes. Two additional targets were used for an active motor thresholding (aMT) procedure before cTBS administration. Coordinates for cortical stimulation of these two sites, the cortical representations of the first dorsal interosseous (FDI) muscle in the right hand, and the tibialis anterior (TA) muscle of the right leg, were also marked in standard space based on prior literature [90,91,92]. All targets were transformed from MNI template (T1) space to each participant’s native (T1) space using the *flirt* and *fnirt* registration tools in the FMIRB Software Library [93]. This allowed for accurate neurostimulation based on optimal TMS coil positioning on the scalp using a frameless stereotaxy system (Brainsight; Rogue Research).
Figure 2cTBS target sites: LBA44 in Broca’s area (aimed at the pars opercularis of the inferior frontal gyrus; MNI coordinates: x = −50, y = 18, z = 18 [52]), MBA8 (MNI coord: x = −6, y = 40, z = 38 [39,52]), and LTOS (MNI coord.: x = −25, y = −85, z = 25 [89]).
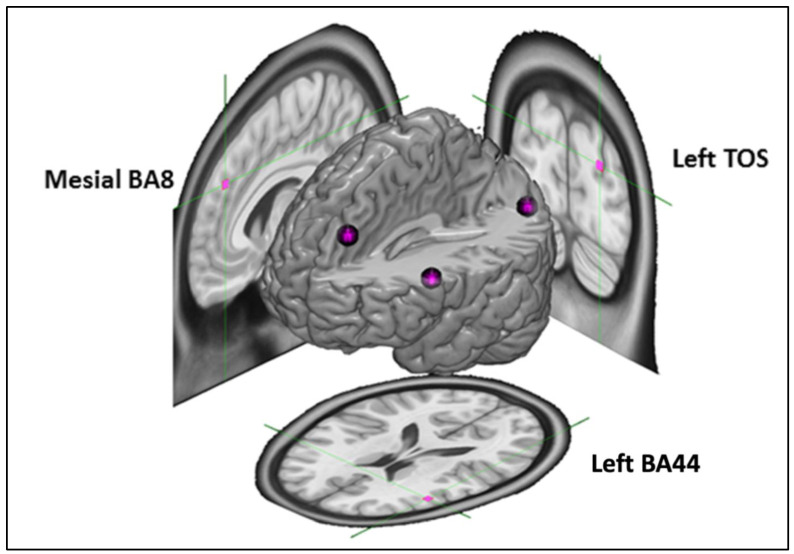



### 2.4. Experimental Sessions

Each participant attended four study visits. The first was a screening visit (no cTBS), which took place in the UCLA Psychology Department, at which the participant was consented and, after viewing one example trial for each task, performed a set of problems analogous to those employed in the subsequent cTBS sessions (except for superficial differences in the stimuli). Participants received no feedback on individual problems or overall performance. To be included in the subsequent cTBS sessions of the study, participants had to perform at or above 50% accuracy on the overall task and each of the three primary subcomponents (i.e., linguistic problems, logic problems, and grammaticality judgments, described below). Seven out of twenty-two recruited participants were excluded for being unable to meet this criterion (five men and two women) while fifteen went on to complete the cTBS phase of the study. The three cTBS sessions took place at the Ahmanson-Lovelace Brain Mapping Center at UCLA. Sessions took place at least one week apart, in order to reduce the likelihood of carryover or saturation effects. In each cTBS session, one of the three sites (LBA44, MBA8, LTOS) was targeted, always in the left cerebral hemisphere, in a counterbalanced fashion. The order in which target sites were stimulated was fully counterbalanced across participants using a Latin square in order to control for order effects. At each visit, participants first performed a ten-minute baseline cognitive task in which all task types were represented. They then underwent the cTBS procedure, which included acquiring active motor threshold (aMT), followed by the administration of cTBS. Approximately two to three minutes after the TMS procedure ended, participants started performing a thirty-minute post-cTBS task. According to prior literature, the inhibitory effects of cTBS are assumed to last 30–60 min [65,94], meaning that the task used in this study should have fallen within the effective window of the procedure. All participants who began the experimental phase of the experiment completed the study. There were no adverse events to report, and all procedures were approved by the UCLA Institutional Review Board.

### 2.5. Continuous Theta Burst Stimulation

For TMS stimulation of the motor cortex representation of the right FDI muscle, LBA44, and LTOS, a Magstim flat figure-eight (double 70 mm) coil was used. Because our MBA8 target and the motor cortex representation of the right TA muscle are located within the interhemispheric fissure, we used an angled figure-eight (double 110 mm) coil that allows better stimulation of deep cortical areas. This method is similar to that used in previous studies [95,96]. After participants completed the baseline task, the aMT was measured for that session’s thresholding target (FDI or TA) using a two-step procedure [97,98]. For LBA44 and LTOS target sessions, the “hot spot” was located near the ‘hand knob’ of the left precentral gyrus [92] (flat figure-eight coil); for MBA8 target sessions, the “hot spot” was searched on the medial wall of the left hemisphere, where the motor representation of leg muscles is typically found (angled figure-eight coil). Single TMS pulses were delivered while the target muscle was mildly activated by having the participant gently squeeze a gel tube while their EMG output was monitored. If single pulses from the coil did not produce motor evoked potentials (MEPs) of 200 μV at the initial location, then the coil location was varied systematically around the initial target site until reliable MEPs were evoked at a suprathreshold intensity. Once the motor cortex “hot spot” was determined, the aMT was determined as the minimum TMS intensity at which motor evoked potentials (MEPs) of at least 200 μV, followed by a silent period, were obtained in at least five out of ten consecutive stimulations under active target muscle contraction.

Following the thresholding procedure, cTBS was applied to the experimental target. In cTBS, triplets of TMS pulses at 50 Hz are delivered at 5 Hz, giving a total of 600 pulses over a period of 40 s. The intensity was set at 80% of the aMT, in accordance with prior studies [65,97,99]. Note that we did not use a depth adjustment, such as that recommended by Stokes [100], because our population consisted primarily of college undergraduates, and cortical atrophy could be expected to be negligible at that age [101].

For 12 out of 45 sessions (5 at which LBA44 was targeted, 1 at which MBA8 was targeted, and 6 at which LTOS was targeted) the participant’s aMT was too high for our TMS device to deliver cTBS without significant heating. For these sessions, instead of using 80% of AMT, we applied cTBS at the highest level allowed by the safety measures of our TMS device (43% of maximum stimulator output (MSO)). The cTBS pulse pattern was generated using a second generation Magstim Rapid2, and the average percentage of MSO used was 35.61% (with a range of 19–43%).

Upon completion of the cTBS stimulation procedure, participants began the post-cTBS task after a delay of approximately two to three minutes. Upon completion of all trials, participants filled out a brief questionnaire to assess how much pain and/or discomfort they experienced during the cTBS stimulation. Both the pain and discomfort scales asked the participant to rate, from 0 to 10, how much pain or discomfort they were in during the procedure, with 0 indicating no pain/discomfort and 10 indicating the worst pain/discomfort they had ever felt.

### 2.6. Analysis

Preprocessing. Response times (RT) for individual trials were checked for outliers greater than three standard deviations above or below the mean, as well as outliers less than one second in duration, which were assumed to be likely accidental button presses given the high task complexity. No >3 SD outliers in either direction were found and four <1 s outliers were removed. We also checked for chance performance in the accuracy data (<50%) for all task types, but no scores below this threshold were found. Accuracy data were prepared for analysis by first averaging across trials for each subject by each combination of task (language, deductive, grammar), site (LBA44, MBA8, L.TOS), and stimulation (pre-cTBS, post-cTBS). Next, the change in accuracy performance between stimulation sessions was calculated as a normalized difference score (i.e., (post–pre)/pre) for each combination of task and site. These accuracy change scores were used as the dependent variable in the subsequent analyses.

The primary model of this study was a 2 (task: language, deductive) × 2 (site: LBA44, MBA8) factorial design, fully crossed and within subjects, which tested the specific predictions of the language-centric and the language-independent views about deductive reasoning and linguistic reasoning, with regard to the change in accuracy from pre to post (i.e., normalized to baseline). We analyzed this model with a 2 × 2 repeated-measures ANOVA followed by pairwise t-tests of simple effects. Because we had specific a priori hypotheses about the direction of the effect, one-tailed tests were used.

Linear trend analysis was used to further investigate relations among tasks and sites using an approach that would allow us to include the control task (grammar) and control site (LTOS) in a statistically valid manner. The first linear contrast was set up as LBA44(−1) vs. LTOS(+0.5) & MBA8(+0.5), and tested whether cTBS inhibition of Broca’s area (LBA44) resulted in decreased accuracy as compared to cTBS inhibition of either MBA8 or LTOS. The second contrast, LBA44(+0.5) vs. LTOS(+0.5) & MBA8(−1), tested whether cTBS inhibition of MBA8 resulted in decreased accuracy as compared to cTBS inhibition of either LBA44 or LTOS. Both of these site contrasts were conducted using all three tasks (language, deductive, grammar), using the dependent variable of accuracy change.

We did not have a specific hypothesis regarding response time, given that participants were instructed to focus on producing accurate answers, and that they were asked to take as long as needed (within the 15 s limit). We have therefore elected not to report an analysis of response times for this study.

## 3. Results

***Descriptive:*Table 2** summarizes the average accuracy before (pre-cTBS) and after (post-cTBS) for each task and stimulation site. All mean accuracies are well above chance, and most are below 90%, indicating that the tasks were difficult but still performed successfully. The logic task appeared to be the most difficult (i.e., had the lowest accuracy). There is a general pattern of performance increase from pre-cTBS to post-cTBS for most combinations of task and site, which may be due to within-participant practice effects over time rather than effects from cTBS stimulation, which are usually inhibitory with protocols like ours [65,67,68]. In contrast to this overall increase in performance, there appears to be decreased linguistic reasoning performance only after stimulation of LBA44 and decreased deductive reasoning performance only after stimulation of MBA8. Normalized difference scores are depicted graphically in Figure 3.

***Factorial ANOVA:*** We tested the specific predictions of the language-centric and language-independent views with a 2 (task: language, logic) × 2 (site: LBA44, MBA8) repeated-measures ANOVA and followed up with two one-tailed *t*-tests, consistent with the predictions of our a priori hypotheses (see Figure 4 for a summary). No main effects of stimulation site (*F*(1, 14) = 0.372, *p* = 0.551, *η*_p_^2^ = 0.026) or task (F(1, 14) = 0.803, *p* = 0.385, *η*_p_^2^ = 0.054) were observed. There was, however, a significant interaction (*F*(1, 14) = 9.299, *p* = 0.009, *η*_p_^2^= 0.399). cTBS inhibition of LBA44 was associated with decreased accuracy in linguistic reasoning, relative to baseline (−5.3%) and increased accuracy in logic, relative to baseline (+12.1%), a difference which was significant (*t*(14) = −2.147, *p* = 0.025 (directional), M_diff_ = −0.1739, effect size *D* = −0.554, 95% confidence interval (CI) = [−0.327, −0.0209], Bayes factor (BF) = 2.984—i.e., a moderate effect). Conversely, cTBS inhibition of MBA8 was associated with increased accuracy in language (+2.8%) and decreased accuracy in logic (−3.6%), but this difference was not significant (*t*(14) = −1.281, p = 0.111, Mdiff = −0.072, 95% CI = [−0.178, 0.034], BF = 0.915—i.e., an anecdotal effect).

***Linear trend analysis.*** We statistically tested the hypothesized relations between all tasks and sites, including controls, with linear trend analyses using site contrasts with the contrast weights encoding for hypothesized accuracy change after cTBS stimulation. See Table 3 for results. Within the linguistic reasoning task, Contrast 1, LBA44(−1) vs. LTOS(+0.5) & MBA8(+0.5), was significant (*F*(1, 14) = 8.207, *p* = 0.012) but Contrast 2, LBA44(+0.5) vs. LTOS(+0.5) & MBA8(−1), was not (*F*(1, 14) = 1.456, *p* = 0.248). Within the deductive reasoning task, Contrast 3, LBA44(−1) vs. LTOS(+0.5) & MBA8(+0.5)—was not significant (*F*(1, 14) = 0.176, *p* = 0.681), but Contrast 4, LBA44(+0.5) vs. LTOS(+0.5) & MBA8(−1), was (F(1, 14) = 10.652, *p* = 0.006). Within the grammatical error task, Contrast 5, LBA44(−1) vs. LTOS(+0.5) & MBA8(+0.5), was significant (F(1, 14) = 8.708, *p* = 0.011) but Contrast 6, LBA44(+0.5) vs. LTOS(+0.5) & MBA8(−1) was not (F(1, 14) = 0.994, *p* = 0.336).

With regard to pain and discomfort, across all participants, the mean pain rating was 2.52 (*SD* = 1.76), while the mean discomfort rating was 3.25 (*SD* = 1.88). For each stimulation site, the mean pain ratings were as follows: 2.64 (*SD* = 1.67) for LBA44, 3.33 (*SD* = 2.09) for MBA8, and 1.60 (*SD* = 0.80) for LTOS. For discomfort ratings at each stimulation site, the means were 3.64 (*SD* = 2.12) for LBA44, 3.87 (*SD* = 1.71) for MBA8, and 2.27 (*SD* = 1.34) for LTOS. No participants who began the TMS component of the study failed to complete it.

## 4. Discussion

A central characteristic of the human mind is the ability to confer meaning to linear, time-dependent, signals by creating abstract, hierarchical, representations of how discrete elements bind to one another [102]. While natural language is a paradigmatic example of this ability [103,104], hierarchical processing characterizes several other aspects of human cognition, including algebra [19,105,106,107], music [22,108,109,110], and action sequences [102,111], among others [11]. Many have thus wondered whether the mechanisms for parsing the structured sequences of language also serve an analogous role in other domains of human cognition [30,31,32].

Here, we addressed this question in the context of the structured sequences of deductive reasoning, and present evidence that is relevant to the hypothesis that, in the adult brain, the structure-dependent operations of logic are parasitic on the mechanisms of language, as suggested by the language-centric view. Specifically, we reported three main results. First, we found support for the dissociability of the syntactic operations of language and the syntax-like operations of deductive reasoning, as evidenced from a significant interaction of the task and site factors. Second, we find that inhibitory cTBS to LBA4, a site traditionally associated with processing the syntax of language [25,26,27,28,29], inhibits performance on the linguistic reasoning task but not on the deductive reasoning task. The presence of this effect, and its directionality, are contrary to the hypothesis that Broca’s area specifically is a common site for processing the structured hierarchies of language and logic, as held by the language-centric view of deduction [32,46,50], and in line with the predictions of the language-independent view [39,42,52,53]. Finally, we failed to find support for the idea that MBA8 plays a core role in deductive reasoning [42] as we expected on the basis of both neuroimaging [39,42,52,53] and neuropsychological [112] data. For, while the pattern of post-cTBS change in the two tasks was consistent with our directional hypothesis (i.e., decreased accuracy in deductive reasoning but not in linguistic reasoning), it was not statistically significant.

It is important to note that while the present experiment is set up in a double-dissociation framework, failure to find the expected effect in medial BA8 does not affect the interpretation of the first two results. That is, the question relating to the dissociability of language and reasoning and, more specifically, whether Broca’s area serves as a site for processing *both* the syntactic operations of natural language and the syntax-like operations of deduction, are fully answered by the first two results. Failure to detect a differential effect in MBA8 only fails to provide support for a specific hypothesis concerning the neural basis of deductive reasoning [39,52,53], despite the neuropsychological data demonstrating that patients with lesions spanning fronto-medial cortices (including MBA8), and no visible damage to Broca’s area, are impaired at deductive reasoning despite ceiling performance on standard neuropsychological tests of language [112].

One potential interpretation of our results is that deduction might be a more distributed process, relying on the “concerted operation of several, functionally distinct, brain areas” [113], and conceivably harder to disrupt with a single-target interference. Indeed, we have previously suggested that “core” deductive processes might be implemented in multiple brain areas, including both the medial BA8 target as well as left rostrolateral prefrontal cortex, in BA10 [39,53]. Furthermore, comparison of the reasoning-impairing lesions reported in the neuropsychological literature (see Figure 1 of ref. [112]) with the known tractography of rostrolateral prefrontal cortex in BA10 (see Figure 5 of ref. [114]) suggests that in the patients enrolled in that study the white matter tracts connecting BA10 with other cortical sites were likely severely damaged. It is thus possible that what caused the deduction-specific impairment in that report was the joint dysfunction of both “core” areas (i.e., medial BA8 and rostrolateral BA10). Unfortunately, our present data cannot speak to the issue, so the plausibility of this hypothesis will require separate empirical assessment.

Although we tested the language-centric hypothesis of deduction in the context of a specific mode of deductive reasoning (i.e., propositional logic), previous work suggests that this conclusion may be expected to also apply to categorical syllogisms [54,73,115], relational problems [116], and pragmatic inferences in the context of naturalistic discourse [117,118]. It should be noted, however, that while our findings imply that the mechanisms of natural language do not participate in the syntax-like operations of deduction, they do not rule out the Vygotskyan idea that language may serve, throughout development, as a “cognitive scaffolding” [9] enabling the acquisition of structure-dependent operations in other domains, such as logic, to then become independent in adulthood. Nonetheless, recent data show that preverbal infants can already demonstrate elementary logic reasoning [119], suggesting that these capacities may actually be distinct even at an early stage of development.

The apparent lack of cTBS effect on three- versus four-variable items is also relevant to two ongoing debates. With respect to logic reasoning, the fact that cTBS to MBA8 impaired equally three- and four-variable logic problems (*t*(14) = −1.19, *p* = 0.127, M_diff_ = −0.07, CI = [−0.21, 0.06]) is contrary to the idea that activity in this region can be explained by non-deductive processes, such as working memory demands imposed by complex deductions [120], or greater relational complexity [87], consistent with recent neuroimaging data [39]. With respect to natural language processing, these results bear on the question of what the precise role of Broca’s area is [25,26,121] and suggest that this region is key to processing the hierarchical, non-local, dependencies of natural language [25,26,27,28,29] and is not just a reflection of verbal working memory load [115]. For, not only does cTBS to this region impair the manipulation of long-distance relationships across non-canonical sentences, but it also fails to differentially affect three- versus four-variable problems (*t*(14) = −0.19, *p* = 0.426, M_diff_ = −0.009, CI = [−0.10, 0.09]), contrary to what a verbal working memory account would predict.

Finally, although extraneous to our ex ante hypotheses, we note that the control site (LTOS) and control condition (grammaticality judgments) offer both some supporting and unexpected observations. On the one hand, the pattern of accuracy change in the control grammar task following cTBS to LBA44 and MBA8 mimics that of linguistic reasoning, as might be expected. On the other hand, cTBS to LTOS did impair performance on grammaticality judgments (but not on either reasoning task), a finding that was entirely unexpected. So, while the control site adds credence to the neurological specificity of the results concerning our two tasks of interest, it poses interpretational challenges, regarding the control site itself, which we cannot address with the present data.

The present study faced several limitations. First, the sample size (*N* = 15) was relatively small. Although this sample size is in line with prior work [80,81,82,83] and it was determined to be adequate by a preliminary power analysis, we acknowledge that verifying the findings will require a larger and more diverse sample and encourage other investigators to replicate and extend the present findings. Second, the target in MBA8 was relatively deep in the cortex, and we did not make a correction to the stimulation strength to account for this (for example, by using the Stokes equation [100]), although we did use a different coil (i.e., angled coil, compared to the flat coil for LBA44 stimulation) that produces deeper stimulation. While this might be an additional explanation for why we found no evidence in support of the third hypothesis, the degree to which this issue might have influenced our results is difficult to assess. It should also be acknowledged that our findings could be the consequence of cTBS facilitating logic reasoning when applied to LBA44 and facilitating linguistic reasoning when applied to MBA8, which would support the language-centric view of deductive reasoning. However, given the prior literature regarding the inhibitory effects of cTBS [65,66,67,68] this explanation seems unlikely.

## 5. Conclusions

In conclusion, the main findings are as follows: (1) the dissociability of the syntactic operations of language and the syntax-like operations of deductive reasoning are sup-ported by the presence of a significant interaction between task and site factors, (2) inhibitory cTBS applied to LBA44 inhibits performance on the linguistic reasoning task but not the deductive reasoning task, (3) we did not find support for MBA8 playing a core role in deductive reasoning as we expected. This work presents preliminary causal evidence that, in the adult healthy brain, abstract logic reasoning can be dissociated from the mechanisms of natural language with non-invasive brain stimulation, a finding that is contrary to the hypothesis that language forms the basis of complex human thought [9,122,123].

## Figures and Tables

**Figure 1 brainsci-13-00067-f001:**
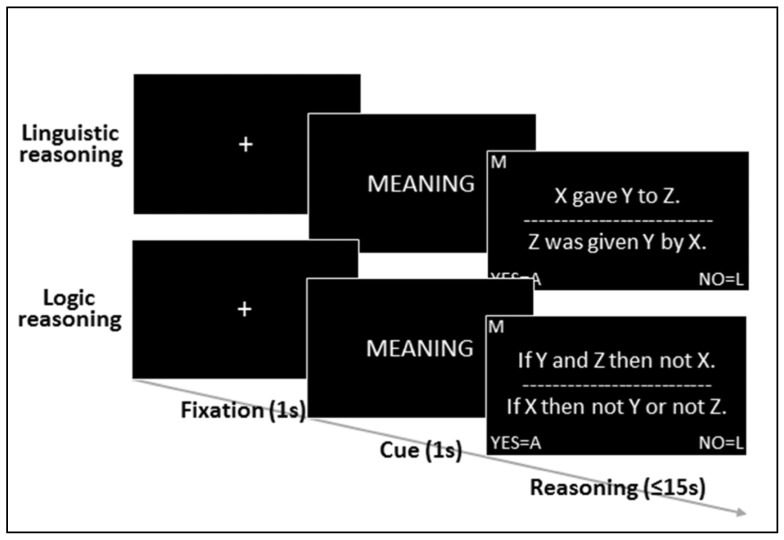
Timeline and sample trials for a linguistic reasoning and a logic reasoning trial (see Table 1 for sample grammaticality judgment trials).

**Figure 3 brainsci-13-00067-f003:**
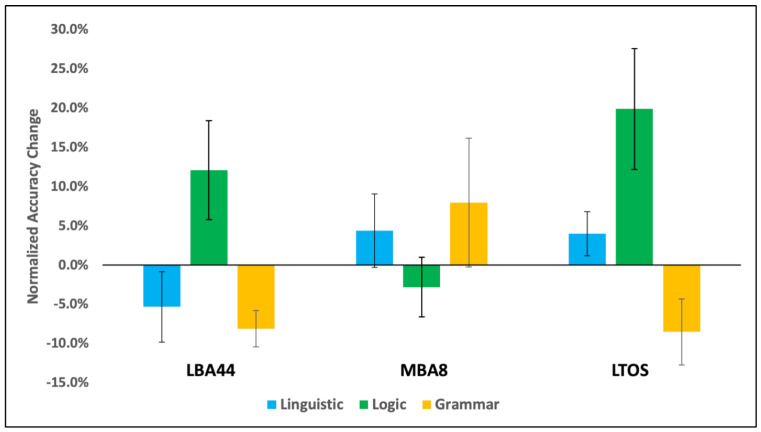
Normalized difference scores. Percent accuracy change for linguistic (blue) and logic (green) reasoning, and grammaticality judgments (yellow), normalized to baseline after cTBS to LBA44 (**left**), MBA8 (**middle**), and LTOS (**right**). Error bars indicate standard error.

**Figure 4 brainsci-13-00067-f004:**
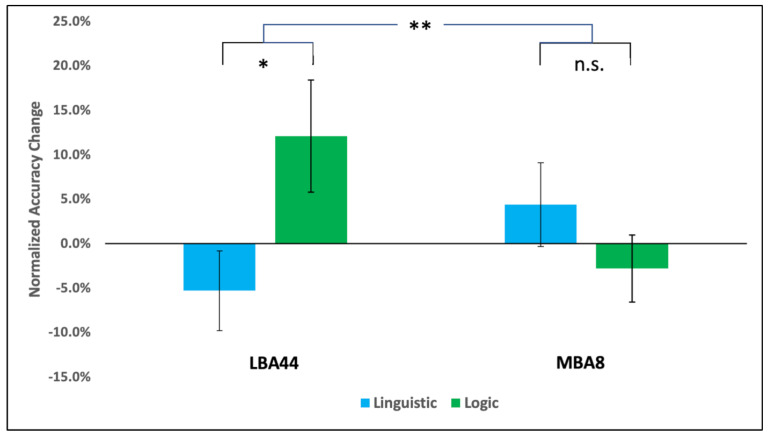
A priori analysis result: Percent accuracy change for linguistic (blue) and logic (green) reasoning after cTBS to Broca’s area (LBA44) (**left**) and MBA8 (**right**) (Error bars indicate standard error; “*” indicates *p* < 0.05; “**” indicates *p* < 0.01, “n.s.” indicates non-significant; see text for details).

**Table 1 brainsci-13-00067-t001:** Example stimuli. Sample logic and linguistic arguments presented in the reasoning and grammaticality judgment tasks (Abbreviations: Log, Logic; Ling, Linguistic).

Reasoning Task			
Type	Terms	**Matching**	**Non-Matching**
Log	3	If both X and Z then not Y.	If either Y or Z then not X.
		If Y then either not X or not Z.	If X then both Y and Z.
Log	4	If both X and not Z then either Y or not W.	If both not Y and not Q then both Z and X.
		If both W and not Y then either Z or not X.	If both Z and X then both not Y and not W.
Ling	3	It was X that Y saw Z take.	It was Y that Z thought X said.
		Z was seen by Y taking X.	Z was thought by Y to have said X.
Ling	4	It was X that W heard Y saw Z take.	What W knew that Y gave Z was X.
		W heard that Z was seen by Y taking X.	It was X that W knew was given to Y by Z.
**Grammaticality Judgment Task**			
Type	Terms	**Grammatical**	**Non-Grammatical**
Log	3	If either Y or X then not Z.	If not Y then Z both and X.
		If Y then either X or Z.	If either not Z or not X then not Y.
Log	4	If either X or W then both Y and Z.	If both Z and not Y then either X or not W.
		If both not Y and not W then both Z and X.	If both W and Y then either not X not or Z.
Ling	3	Z was thought by Y to have said X.	It was to Y that from Z told X.
		It was Y that X thought Z said.	What Z told Y was X.
Ling	4	Z knows that X is given by Y to W.	Z will be seen by Y taking X is what W will hear.
		If either W or X then both Y and not Z.	It was X that W heard Y take Z saw.

Note: The disjunctions in these stimuli relied on De Morgan’s Laws, and that participants were instructed to treat the “or” connective as an “inclusive or” for purposes of this study (see text).

**Table 2 brainsci-13-00067-t002:** Mean accuracy for each task before (Pre-cTBS) and after (Post-cTBS) transient inhibitory stimulation to each site.

	Stimulation Site					
	LBA44		MBA8		LTOS	
	Pre-cTBS	Post-cTBS	Pre-cTBS	Post-cTBS	Pre-cTBS	Post-cTBS
Linguistic Reasoning	94%	87%	82%	85%	85%	87%
Logic Reasoning	72%	76%	78%	75%	69%	78%
Grammaticality Judgment	90%	83%	83%	85%	80%	85%

**Table 3 brainsci-13-00067-t003:** Linear trend analysis.

Contrast	Task	Stim. Site & Contrast Weights	*F*	Signature
1	Linguistic	LBA44(−1) vs. LTOS(+0.5) & MBA8(+0.5)	8.207	**0.012 ***
2		LBA44(+0.5) & LTOS(+0.5) vs. MBA8(−1)	1.456	0.248
3	Logic	LBA44(−1) vs. LTOS(+0.5) & MBA8(+0.5)	0.176	0.681
4		LBA44(+0.5) & LTOS(+0.5) vs. MBA8(−1)	10.652	**0.006 ***
5	Grammar	LBA44(−1) vs. LTOS(+0.5) & MBA8(+0.5)	8.708	**0.011 ***
6		LBA44(+0.5) & LTOS(+0.5) vs. MBA8(−1)	0.994	0.336

Note: For each task, contrast weights per stimulation site are given, followed by F value and significance. Significant contrasts highlighted in **bold**. Those contrasts which survive a Bonferroni correction are marked with a *.

## Data Availability

Study data available at the Open Science Foundation: https://osf.io/3yh2q/ (accessed on 2 November 2022).

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
