# Peer review of "Dissociating Language and Thought in Human Reasoningâ€"

_brainsci, 2022, doi:10.3390/brainsci13010067_

Round 1
Reviewer 1 Report
This is a very well-designed study examining the relation between deductive reasoning and linguistic process. The paper is very well-written and easy to understand. The methodology is strong too. I believe that the paper will have important implication to the literature. I only have one comment which is related to their power analysis. In the method section, it will be important for the authors to justify further the expected effect size that is chosen for the power analysis.
Author Response
RESPONSE:
We thank the reviewer for the helpful and insightful feedback. We agree that further clarification regarding the effect size would improve our manuscript. We have added the following language at lines 136-146:
A total of 45 cTBS sessions were obtained from 15 participants in a three-session within-subjects design. All study procedures took place at the University of California, Los Angeles (UCLA). A power analysis was conducted with G*Power [66] using an effect size from a factorial ANOVA interaction in a previous study [60]. In that study by Tsujii et al., inhibitory rTMS (1 Hz) applied to left IFG (BA45) impaired reasoning about congruent but not incongruent syllogisms, thereby eliminating the belief bias, which is the bias that reasoners have for judging syllogisms as valid when the conclusion agrees with their preexisitng knowledge. Tsujii et al. reported a conventionally “large” effect size (η2p=.13) for the effect of IFG stimulation on congruent reasoning. We further used an assumed α=0.05, and determined that 15 participants would be needed to achieve 85% power to detect an interaction using a repeated measures ANOVA.
Reviewer 2 Report
The paper is well written and relatively interesting.
However, because of the relevance of its topic and its intrinsic value, it should be better formatted and structured, and this, I think, requires a revision.
Section 1, Dissociating language and thought in human reasoning, which is an introduction, should be structured, indeed, as a proper introduction (perhaps starting from the title) and expanded at least by stressing on the relevance of the paper in itself and in its field of studies and explaining a little better / more comprehensively the aim and scope of the research. Therefore, simply, expand, which means, almost automatically, enhance.
The literature review is there and is not there, in the meaning that it is 'scattered' all over the paper. Why not to implement a specific section, entitled (originally) "Literature Review", after the Introduction and before the section on methodology? This would allow for a clearer survey and analysis of the works used and cited by the Authors and would provide the readers, even a non-specialized audience, with a more solid and 'user-friendly' outline of relevant works in the field. I think a literature review properly organized and easily readable is necessary, in a paper like this (in all the papers, after all).
The sample for analysis can be an 'item' for discussion, in the ideal 'check-list' of the paper. Is the sample significant enough? Should it be 'bigger' / more structured? I do not know, honestly - in my opinion, it can be considered indicative, but not consistent enough for an exemplary study.
Besides this, it is necessary to see how the Authors work on the sample, and they work quite well.
The experiment is ok, in my opinion, the results are effectively delineated, despite the related section could be expanded a little, for the sake of comprehensiveness and clarity.
The Discussion should be definitely expanded, adding more analysis and comments, which, after all, should be the substance and the 'meat' of the paper - moreover, some elements from the Methods section, which is extensive, could be 'exported' into the Discussion section (I mean, discussed there, literally).
The Conclusions are a bit of a 'joke'. I mean, often I find the Conclusions, in a paper, futile, but, since we need them, we need them, therefore the length of the Conclusions of this paper is not acceptable at all. The section needs to be expanded, with more remarks and (final) observations, and also by 'mirroring' the introduction, highlighting again the most significant research goals of this study and how the Authors have achieved them.
Nothing to say about the English language, very good and professional.
To summarize, this paper is quite interesting and, in a way, significant. However, it gives a lot of things (too many) for granted, like if it were not necessary to include them in the article (while it is), and shows a 'messy' format.
Because of this, it requires a thorough revision, which surely will improve it considerably.
Thank you very much.
Author Response
RESPONSE:
We thank the reviewer for their thoughtful comments. We agree that more information about the existing literature would make the paper more accessible for those readers who may be less familiar with our field. To accomplish that we have modified the language at lines 52 - 68 as follows:
Under the first view, which we will call here the language centric-view, the syntax-like operations of deductive reasoning are mainly based upon the neural mechanisms of language in the left IFG [33,34], and thus best understood as linguistic in nature. This view rests, neuroscientifically, on a number of publications showing activation of the left IFG, as detected with functional MRI, during logic inferences [33–38]. Reverberi and colleagues [33], for example, used an event-related fMRI approach to probe different cognitive components of the inferential process and found that simple conditional and disjunctive problems (i.e., based on logic connectives such as “if … then”, “or”) re-cruited, at the moment of inference, the left IFG (in BA44) along with other lateral frontal (i.e., BA6) and parietal (BA40) regions. Furthermore, these findings were shown to apply not only to conditional and disjunctive propositional problems but also to syllogistic in-ferences (i.e., based on quantifiers such as “all”, “some”, “none”) [34]. Indeed, Reverberi and colleagues also showed that BA44 and BA45 in the left IFG were recruited during both the encoding and premise integration stages of a deductive inference, which was inter-preted by the authors as suggesting a central role for these primarily linguistic regions in deductive reasoning (See [29] and [31] for a review).
And at lines 69-89 as follows:
Under the second view, which we will call here the language-independent view, deductive reasoning is mainly supported by neural mechanisms that extend beyond the areas of the brain conventionally associated with processing the hierarchical relationships of language [39], spanning left dorsomedial frontal and frontopolar cortices (in Brodmann areas (BA) 8 and 10, respectively) [26,27,40–46,46,47]. This view is also primarily sup-ported, in the neuroscientific literature, by functional MRI work. Coetzee et al. [26], for example, used an event related fMRI approach to probe neural activation during a conditional reasoning task and found that areas in the IFG were selectively activated by increased (non-logic) verbal processing demands, but not by increased deductive rea-soning demands. Conversely, medial frontal cortex was specifically activated by in-creased deductive reasoning load, but not by verbal non-deductive load. This study was in part a replication of earlier work by Monti et al. [40], in which the comparison of brain activations during difficult versus easy deductive inferences highlighted a non-linguistic network subserving deductive reasoning, interpreted to be composed of core deductive regions (in frontopolar BA10 and frontomedial BA8) and general cognitive support re-gions (including left frontal BA6 & 47, and parietal (BA7 & 40)) that are more flexibly recruited depending on working memory and other ancillary cognitive processes. Similar findings were reported in a decidedly different study design by Rodriguez-Moreno et al. [41]. In this work, Rodriguez-Moreno and colleagues found that syllogisms presented in either a visual or auditory manner revealed a similar pattern of brain regions to the studies mentioned above, with a similar absence of recruitment of classical language areas.
We have also taken the reviewer’s advice and expanded the conclusion as follows:
In conclusion, the main findings are as follows: 1) the dissociability of the syntactic operations of language and the syntax-like operations of deductive reasoning are sup-ported by the presence of a significant interaction between task and site factors, 2) in-hibitory cTBS applied to LBA4 inhibits performance on the linguistic reasoning task but not the deductive reasoning task, 3) we did not find support for MBA8 playing a core role in deductive reasoning as we expected. This work presents preliminary causal evidence that, in the adult healthy brain, abstract logic reasoning can be dissociated from the mechanisms of natural language with non-invasive brain stimulation, a finding that is contrary to the hypothesis that language forms the basis of complex human thought [108,112,113].
Reviewer 3 Report
this study is based on only 15 participants. Authors may be invited to resubmit their work if the number of participants is sufficient. A minimum of 18-21 participants is required for linguistic studies.
Author Response
RESPONSE
We thank the reviewer for their attention to the issue of sample size. We agree about its importance. While it is unfortunately no longer possible for us to add additional participants to this study, since some of the original researchers have moved to other institutions, we believe there are good reasons to assume that this sample size is sufficient for the paper currently under consideration. The first is that we conducted an a priori power analysis using an effect size from a previously published and similar neuromodulation study, and determined we had 85% power to detect a statistical interaction with a repeated measures ANOVA. The second is that samples in this range are not unusual in early/exploratory neuromodulation studies such as this one. To support both of these points we have added the following text on lines 138-148.
A power analysis was conducted with G*Power [66] using an effect size from a factorial ANOVA interaction in a previous study [60]. In that study by Tsujii et al., inhibitory rTMS (1 Hz) applied to left IFG (BA45) impaired reasoning about congruent but not incongruent syllogisms, thereby eliminating the belief bias, which is the bias that reasoners have for judging syllogisms as valid when the conclusion agrees with their preexisitng knowledge. Tsujii et al. reported a conventionally “large” effect size (η2p=.13) for the effect of IFG stimulation on congruent reasoning. We further used an assumed α=0.05, and determined that 15 participants would be needed to achieve 85% power to detect an interaction using a repeated measures ANOVA. This sample size is consistent with other previously published rTMS work involving the role of the IFG in cognition [67–70].
We have also been careful throughout the paper to label our findings as preliminary, and to be cautious regarding the strength of our claims.
While we can no longer add participants to our study, we sincerely hope that publishing our findings will make it more likely for other researchers to attempt a much needed replication in a larger sample, thereby helping to add more clarity to the important questions being addressed here about human cognition.
Round 2
Reviewer 2 Report
The paper has been improved in critical parts.
However, the Authors would have been able to enhance it further, especially at the level of literature review.
Despite this, the paper is good in itself, and, now, can be considered for publication.
Thank you and regards.
Author Response
Response:
We thank the reviewer for their incisive feedback. We have added additional material to the literature review on lines 46-56 to make it more complete.
“The antagonism between these two polar positions has long been investigated in the context of a number of cognitive domains including the development of ontological categories [13], color perception [14,15], spatial cognition and geometry [16,17], number cognition [18,19] , action representation [20], music cognition [21,22], and theory of mind [23,24], among many others. More recently, this debate has also leveraged the tools of neuroscience to investigate the degree to which the neural substrate of natural language also participates in other aspects of human cognition [11]. In particular, it has been proposed that a segment of the left inferior frontal gyrus (IFG; often referred to as Broca’s Area), which has long been associated with processing the hierarchical sequences of natural language [25–29], might actually function supramodally as a processor of hierarchical sequences across domains of human thought [30–32].”
We hope the literature review is now satisfactory. Thank you.
Reviewer 3 Report
Dear Authors,
Your explanations sound reasonable.
Thank you for revising the manuscript.
Kind regards,
Author Response
We appreciate the author's feedback, and appreciate their contribution to our manuscript.